# SLIDING-WINDOW ATTENTION FOR REINFORCED REASONING

## ABSTRACT

Large reasoning models such as DeepSeek-R1 employ reinforcement learning (RL) to incentivize the reasoning capability. As the context length growth, the quadratic complexity of `self-attention (SA)` prohibits scaling to longer contexts. Recently, hybrid, sparse and linear attention methods aim to reduce the cost of SA, yet suffer from costly retraining, high complexity or linear memory growth. To address it, we revisit `sliding-window attention (SWA)`. It not only offers linear-time complexity and constant memory, enabling faster RL rollouts, but also facilitates cheap conversion from pretrained transformers. Notably, we prove that SWA can handle the reasoning tasks well due to the locality of thought. In this paper, we introduce **Sliding Window Attention for Reinforced Reasoning (SWARR)**, a two-stage approach: (1) math-specific supervised fine-tuning to convert a pretrained `SA` model into a `SWA` as cold-start, and (2) RL optimization using DAPO (Yu et al., 2025a) to enhance reasoning capabilities. Under same settings, our *SWARR* outperforms `SA` by 1.78% on 1.5B, while delivering $6.2\times$ higher throughput and $8\times$ larger batch size, and $1.5\times$ longer context under same memory budget. Our *SWARR* achieves the competitive performance among 1.5B and 7B models, surpassing the DeepSeek-R1-Distill-Qwen-1.5B and 7B by 1.9% and 3.4% respectively. To our knowledge, this is the first work to show that trained `SWA` is a competitive alternative to transformers, enabling efficient and scalable reasoning.

## 1 INTRODUCTION

Reinforcement Learning (RL) (Shao et al., 2024; Yu et al., 2025a; Schulman et al., 2017; Ouyang et al., 2022) offers a promising path for advancing the chain-of-thought (CoT) (Wei et al., 2022) reasoning (DeepSeek-AI, 2025; Zeng et al., 2025; Team et al., 2025a; Bai et al., 2025) in Large Language Models (LLMs). However, transformer-based LLMs face significant challenges when processing long sequences due to the quadratic complexity of `self-attention (SA)` (Vaswani et al., 2017; Dao et al., 2022). As the sequence length grows, the SA-based models are constrained by the quadratic increase in computation complexity and linear escalation in memory usage, making it hard to scale to longer contexts during RL rollouts.

To address this, prior works explored three major families of efficient attention mechanisms. Hybrid attention (Wang et al., 2025b;a; OpenAI, 2025), as in MiniMax-M1 (Chen et al., 2025a) and Nemotron-H (Blakeman et al., 2025), interleaves `SA` layers with linear attention like Mamba2 (Gu & Dao, 2023; Dao & Gu, 2024) or Lightning Attention (Qin et al., 2024a). While this reduces the fraction of quadratic-cost layers, overall complexity remains bounded by the $O(L^2)$ components. Besides, hybrid attention typically requires costly training from scratch. Sparse attention (Yuan et al., 2025; Lu et al., 2025; Xiao et al., 2024) like NSA (Yuan et al., 2025) and MOBA (Lu et al., 2025) reduce the quadratic cost of `SA` by restricting each query to a subset of keys, but they necessitate training from scratch. Despite these advantages, linear attention models are harder to obtain, particularly for reasoning. Broadly, there are two approaches: (1) training from scratch, which is costly and time-consuming; and (2) converting pretrained `SA` models into linear-attention variants using Liger (Lan et al., 2025), which requires a post-training process to convert SA to GLA (Yang et al., 2023). This raises a question: ***How to retain the strengths of self-attention for reasoning while making it practical and efficient for RL training?***

For this questions, we revisit the `Sliding-Window Attention (SWA)` (Beltagy et al., 2020; OpenAI, 2025) with three key observations: ① **`SWA` can retain the reasoning ability of `SA` for Reasoning Tasks due to Locality of Thought:** Recent studies (Prystawski et al., 2023) reveal the "locality of thought" phenomenon, which suggests that attention is heavily concentrated on recently generated tokens in CoT, indicating that `SWA` can effectively handle reasoning tasks. We further validate it by visualizing the cumulative distribution function (CDF) of attention scores of `SA` model on reasoning tasks in Figure 1 and find that around 80% of attention mass is concentrated within a 4k token window (refer to Appendix D for details). This locality bias makes `SWA` effective for complex reasoning. Besides, As illustrated in Figure 2,

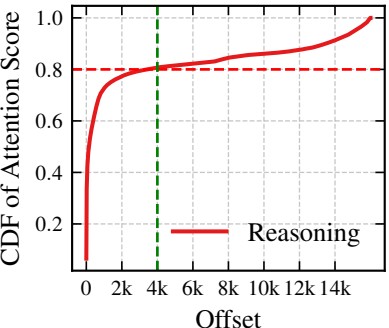

Figure 1: CDF of attention scores: 80% mass within 4k tokens, showing locality of thought.

stacking `SWA` layers expands the effective receptive field far beyond the individual window size, enabling the model to capture long-range dependencies. ② **`SWA` is Efficient in RL Training:** The constant memory footprint of `SWA` leads to significantly reduced memory usage relative to the linear memory growth of `SA`. `SWA` facilitates larger rollout batch sizes or larger group sizes during RL training under the same memory budget, which in turn allows for more effective policy gradient estimation. As presented in Sec. 4.3, `SWA` outperforms `SA` under equal RL training time budget, where we can exploit the memory savings to increase batch size, group size or context length. ③ **`SWA` Enables Practical Conversion from Pretrained `SA` Models:** Linear attention requires extensive retraining or post-training to adapt from pretrained `SA` models. We choose `SWA` because it retains the fundamental structure of `SA` but confines attention to a local window. It naturally align with SA. We compare the Liger (Lan et al., 2025) with `SA` and `SWA` during SFT stage and find that Liger is slower to converge as shown in Figure 9. Additionally, `SWA` is compatible with highly optimized kernels such as FlashAttention (Dao et al., 2022), benefiting from mature, production-grade GPU acceleration. These observations motivate us to revisit `sliding-window attention (SWA)` (Beltagy et al., 2020) for efficient RL reasoning.

Based on the above observations, we propose **Sliding Window Attention for Reinforced Reasoning (SWARR)** framework, as illustrated in Figure 3. *SWARR* consists of two stages: (1) a supervised fine-tuning (SFT) stage that adapts a pretrained `SA` into `SWA` for cold-start initialization. For SA model, SFT extends the context length (e.g., from 4k to 32k), while for SWA, it enables efficient conversion. (2) Reinforcement learning optimization using DAPO (Yu et al., 2025a), which further enhances the reasoning capabilities of the `SWA` policy.

Extensive experiments on math reasoning benchmarks demonstrate that our *SWARR* outperforms `SA` baselines by 1.78% on 1.5B, while delivering significantly faster rollouts and supporting longer effective contexts under the same training time. Our *SWARR* achieves the competitive performance among 1.5B and 7B models, surpassing the DeepSeek-R1-Distill-Qwen-1.5B and 7B by 1.9% and 3.4% respectively. During training, `SWA` achieves 1.23× faster for SFT. Under constrains of 65G memory, `SWA` achieve 2.7× higher thoughput and 8× larger batch size during RL rollout. To our knowledge, this is the first work to systematically demonstrate that carefully trained `SWA` backbones can serve as competitive alternative to SA.

Our contributions are as follows:

- We propose **SWARR**, a two-stage framework that successfully adapts a pretrained Self-Attention (SA) model to a Sliding-Window Attention (SWA) architecture for efficient reinforced reasoning. We are the first to systematically demonstrate that `SWA` can serve as a competitive and highly efficient alternative to `SA` for complex reasoning tasks.
- We introduce a practical and effective methodology for converting `SA` models to SWA, consisting of a math-specific supervised fine-tuning (SFT) stage for cold-start adaptation, followed by RL with DAPO to unlock advanced reasoning capabilities.
- We demonstrate SWA achieves better performance and efficiency than SA. Our 1.5B *SWARR* model outperforms its `SA` counterpart by 1.78% on key benchmarks while achieving 6.2× higher throughput and supporting 8× larger batch sizes. Furthermore, our *SWARR* model surpasses DeepSeek-R1-Distill-Qwen by 1.9% and 3.4% at 1.5B and 7B scales respectively.

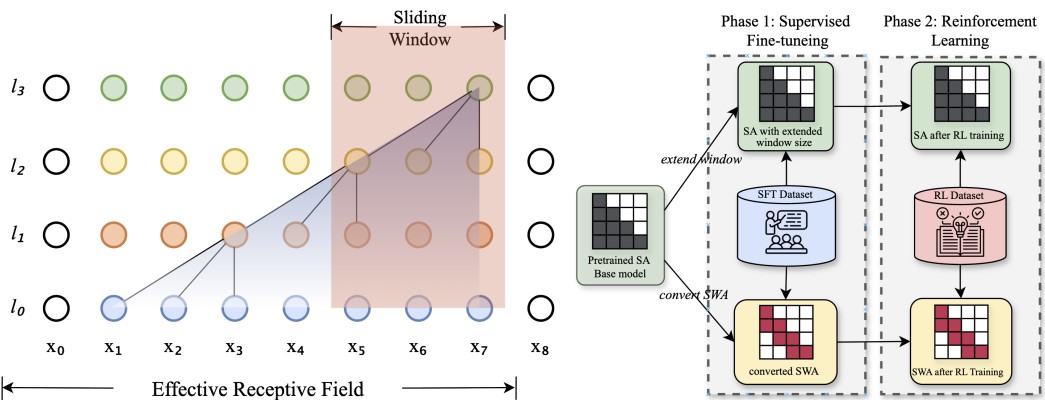

Figure 2: Sliding-window attention: each layer attends locally, but stacking layers expands the effective receptive field.

Figure 3: Overview of the *SWARR* pipeline. For SA, SFT extends the context length; for SWA, SFT converts SA to SWA.

## 2 RELATED WORK

**Efficient Attention Mechanisms.** SA (Vaswani et al., 2017) offers a global receptive field but at a prohibitive $O(L^2)$ computational cost. We group efficient variants by the conversion from pretrained SA as shown in Table 1: sparse attention (Yuan et al., 2025; Lu et al., 2025; Xiao et al., 2024) restricts keys via fixed, data-independent patterns and typically requires training from scratch; linear attention (Choromanski et al., 2020; Peng et al., 2025; Glaeser, 1999) replaces softmax with kernel mappings $\phi(\cdot)$ to achieve $O(L)$, but this operator change complicates direct conversion and often demands post-training procedures (Lan et al., 2025); in contrast, sliding-window attention (SWA) (Beltagy et al., 2020) retains the original softmax operator and parameterization, achieving $O(wL)$ by masking to a local window of size $w$. This compatibility makes SWA a practical, near drop-in path for adapting pretrained SA backbones.

Table 1: Comparison of efficient attention mechanisms.

| Feature | Self-Attention (SA) | Sparse Attention | Linear Attention | Sliding-Window (SWA) |
|---|---|---|---|---|
| **Formulation** | $\mathrm{softmax}\left(\frac{QK^\top}{\sqrt{d_k}}\right)V$ | $\mathrm{softmax}_{j \in S_t}\left(\frac{q_t k_j^\top}{\sqrt{d_k}}\right)v_j$ | $\phi(Q)(\phi(K)^\top V)$ | $\mathrm{softmax}\left(\frac{Q_w K_w^\top}{\sqrt{d_k}}\right)V_w$ |
| **Complexity** | $O(L^2)$ | Sub-quadratic | $O(L)$ | $O(wL)$ |
| **Conversion** | Baseline | Requires new sparsity patterns | Requires operator redesign | Compatible with fine-tuning |
| **Key Property** | Global receptive field | Fixed sparse patterns | Associative property | Local receptive field |

**Self-attention LLMs with RL.** Most reasoning models are trained with a self-attention backbone. Recent work has shown that RL can substantially improve the reasoning capabilities of large language models. Notably, DeepSeek-R1 (DeepSeek-AI, 2025) uses GRPO (Shao et al., 2024) for RL training on a self-attention model and reports strong gains on reasoning benchmarks. Following GRPO, methods such as DAPO (Yu et al., 2025b), GRESO (Zheng et al., 2025), SimpleRL-Zoo (Zeng et al., 2025), and Seed-GRPO (Chen et al., 2025b) have been introduced to further boost performance. However, due to the limitations of self-attention, RL scaling still suffers from quadratic computation, limiting scalability for long-context rollouts.

**Hybrid Attention LLMs with RL.** Recent work has introduced hybrid attention models to improve RL rollout efficiency by combining local or selective attention with standard SA. For example, MiniMax-M1 (Chen et al., 2025a) uses a 7:1 interleaving of Lightning Attention (Qin et al., 2024a) and standard SA, optimized with the CISPO (Chen et al., 2025a) RL algorithm, to achieve improved RL rollout efficiency and enhanced performance in sequential decision-making tasks. GPT-OSS (OpenAI, 2025) uses a 1:1 interleaving of SWA with an attention sink (Xiao et al., 2023) and standard SA to achieve efficient long-context processing and improved streaming performance in large-scale language modeling. Although these hybrid designs can partially alleviate KV cache growth and reduce computation, they do not fully escape the quadratic complexity inherent in the

`SA` components. This persistent bottleneck motivates the exploration of architectures with purely linear complexity that can completely eliminate the quadratic scaling issue.

**State-Space LLMs with RL.** Beyond self-attention variants, state-space models (SSMs) such as S4/S5 (Gu et al., 2021), Mamba (Gu & Dao, 2023), and Mamba2 (Dao & Gu, 2024) have emerged as promising alternatives for handling long sequences with linear complexity. A prevalent trend involves creating hybrid Mamba-Transformer architectures. However, many of these models necessitate extensive training from scratch. For instance, Nemotron-H (Blakeman et al., 2025), Hunyuan-Turbo (Team et al., 2025b), and Nemotron-Nano-2 (Basant et al., 2025) all employ hybrid designs that require significant computational investment. Other approaches, like Jamba (Lieber et al., 2024), use an interleaved Transformer-Mamba mixture-of-experts architecture, while Jet-Nemotron (Gu et al., 2025) utilizes neural architecture search to find an optimal combination. More recently, M1 (Wang et al., 2025b) has aimed to reduce this training burden by first distilling a Transformer into a hybrid architecture before applying SFT and RL. In contrast to these methods, our `SWA` offers a much simpler conversion path, as it does not require a complex distillation process, making it a more practical alternative for adapting pretrained models.

## 3 SLIDING-WINDOW ATTENTION FOR REINFORCED REASONING (SWARR)

This section outlines a two-stage methodology for SWARR, which transforms a pretrained transformer into a model capable of high-performance reasoning under a fixed computational budget.

### 3.1 STAGE 1: SUPERVISED FINE-TUNING

The first stage converts an `SA` model to `SWA` as a cold-start. We initialize the weights of the new `SWA` model directly from a pretrained transformer, reusing the existing parameters of its query, key, value (QKV), and multi-layer perceptron (MLP) layers. The primary architectural modification is the replacement of the global attention mechanism with a localized, sliding-window one, which constrains the receptive field of each token to a fixed number of its immediate neighbors.

This shift in inductive bias necessitates adaptation. To ensure the model learns to reason effectively within its new limited context length, we perform supervised fine-tuning (SFT) on general mathematical datasets (Liu et al., 2025c). The objective function is the standard cross-entropy loss over the target sequences:

$$\mathcal{L}_{\text{SFT}}(\theta) = \mathbb{E}_{(x,y)\sim\mathcal{D}_{\text{SFT}}}\left[-\sum_{t=1}^{L}\log\pi_{\text{SWA}}(y_t|x, y_{<t})\right] \tag{1}$$

This process acclimates the model to its localized attention pattern and recovers any performance degradation resulting from the altered attention pattern. For fair comparison, we also conduct SFT for the `SA` model to extend its context length (e.g., from 4k to 32k) using the same dataset and training settings, following the pipeline in Figure 3.

### 3.2 STAGE 2: REINFORCED REASONING

To further enhance the reasoning abilities of SWA, we employ reinforcement learning (RL) using the DAPO (Yu et al., 2025a) recipe. Specifically, we use dynamic sampling to filter out uninformative sample groups, apply token-level policy-gradient updates, and use reward shaping to handle truncated sequences. The optimization is guided by the following objective:

$$\mathcal{L}_{\text{DAPO}}(\theta) = \mathbb{E}_{(q,a)\sim\mathcal{D},\{o_i\}_{i=1}^{G}\sim\pi_{\theta_{\text{old}}}(\cdot|q)}\left[\frac{1}{\sum_{i=1}^{G}|o_i|}\sum_{i=1}^{G}\sum_{t=1}^{|o_i|}\frac{\pi_\theta(o_{i,t}\mid q, o_{i,<t})}{\pi_{\theta_{\text{old}}}(o_{i,t}\mid q, o_{i,<t})}\hat{A}_{i,t}\right], \tag{2}$$

where the fraction is the importance-sampling ratio and $\hat{A}_{i,t}$ denotes the group-normalized advantage. To focus the optimization, updates are performed only on groups with meaningful reward signals, and KL regularization is omitted to prioritize reward maximization. We append the prompt "`Please reason step by step, and put your final answer within \boxed{}.`" to each question to elicit chain-of-thought reasoning. The reward is computed from the final-answer correctness. For more details of RL, please refer to Appendix E.4.

Table 2: Comparison of RL reasoning models at 1.5B scale on reasoning benchmarks. All models use Qwen2.5-Math-1.5B as the base. $^*$ indicates results measured by us. "-" denotes not available.

| Model | AIME24 | Math500 | AMC | Olymp | Average |
|---|---|---|---|---|---|
| *1.5B-scale SA models* | | | | | |
| Qwen2.5-Math-1.5B (Yang et al., 2024) | 3.3 | 35.6 | 34.7 | 21.6 | 23.8 |
| DeepSeek-R1-Qwen-1.5B$^*$ (DeepSeek-AI, 2025) | **27.7** | 81.2 | **62.8** | 52.9 | 56.2 |
| GRESO-1.5B (DM) (Zheng et al., 2025) | 15.0 | 76.6 | 61.4 | 38.5 | 47.9 |
| GRESO-1.5B (OR1) (Zheng et al., 2025) | 20.0 | 76.1 | 50.6 | 39.2 | 46.5 |
| RLSC (Li et al., 2025) | 6.7 | 62.4 | 46.2 | 29.9 | 36.3 |
| Oat-Zero-1.5B (Liu et al., 2025b) | 20.0 | 74.2 | 53.0 | 37.6 | 46.2 |
| TTRL (Zuo et al., 2025) | 15.8 | 73.0 | 48.9 | - | 45.9 |
| TAPO (Wu et al., 2025) | 16.7 | 69.0 | 55.0 | 33.6 | 43.6 |
| Seed-GRPO (Chen et al., 2025b) | 23.3 | 75.4 | 50.6 | 41.3 | 47.7 |
| EDGE-GRPO (Zhang et al., 2025) | 10.0 | 73.2 | 44.6 | 37.3 | 41.3 |
| *1.5B-scale Hybrid/SWA models* | | | | | |
| M1-3B (Wang et al., 2025b) | 29.0 | 82.1 | **62.8** | 47.3 | 55.3 |
| SWARR-1.5B (Ours) | **27.7** | **87.0** | 61.4 | **56.1** | **58.1** |

## 3.3 INFRASTRUCTURE

Our infrastructure is built on XTuner (Contributors, 2023) and employs Fully Sharded Data Parallel (FSDP) (Zhao et al., 2023) for memory-efficient distributed RL training. To accelerate the critical rollout phase, we adopt a colocation strategy similar to HybridFlow (Sheng et al., 2024), allowing training and inference on the same devices to eliminate weight-conversion overhead. The inference engine is further optimized with a centralized request distributor, continuous batching (Yu & Jeong, 2022), and asynchronous rollout (Fu et al., 2025) to maximize throughput and resource utilization.

Moreover, our infrastructure is tailored to `SWA` architectural properties. Unlike `SA`, which requires complex KV-cache management (Kwon et al., 2023), `SWA`'s fixed-size attention window enables a more efficient caching mechanism. We implement a ring-based buffer for the KV cache, illustrated in Figure 11, that maintains a fixed-size memory buffer, overwriting the oldest key–value pairs circularly once the window capacity is reached.

## 4 EXPERIMENTS

### 4.1 EXPERIMENTAL SETUP

**Model.** Experiments are conducted on Qwen2.5-Math models at 1.5B and 7B scales. `SA` models are fine-tuned to support extended context length from 4k to 32k tokens. All models are initialized from pretrained Qwen2.5-Math model with 4k context length. `SWA` models are converted from `SA` models with window sizes of 2k, 4k, and 8k. We denote these models as `SWA-2k`, `SWA-4k`, and `SWA-8k` respectively. Without specifically noted, we use `SWA-4k` as the default *SWARR* model.

**Datasets.** For SFT, we use the AceReason-1.1-SFT dataset (Liu et al., 2025c), which contains 2,668,741 math and 1,301,591 code samples. The 1.5B models are trained on the math subset (19B tokens), while the 7B models are trained on both math and code subsets (58B tokens). For RL training, we utilize the AceReason-Math dataset (Liu et al., 2025c), containing 49,000 challenging math problems designed to rigorously stimulate the reasoning abilities.

**Evaluation.** For evaluating mathematical reasoning models, we assess performance on a suite of competition-level benchmarks: MATH500 (Hendrycks et al., 2021), AIME24 (MAA, 2025), AMC (MAA, 2023), and OlympiadBench (He et al., 2024). Each benchmark is repeated 2, 32, 8, and 2 times respectively, resulting in 1000, 960, 664, and 1348 samples to ensure robust and reliable evaluation. These datasets collectively measure the models' ability to solve complex mathematical reasoning problems and provide a comprehensive assessment of reasoning proficiency.

**Training Settings.** We present two settings. **(1) Fair comparison training settings:** During the SFT stage, the learning rate is initialized at $7 \times 10^{-6}$ and decayed to $7 \times 10^{-7}$ using a cosine annealing schedule. Weight decay is set to 0.1. SFT is performed on 4 H800 nodes (32 GPUs total).

Table 3: Comparison of RL reasoning models at 7B scale on reasoning benchmarks. All models use Qwen2.5-Math-7B as the base. $*$ indicates results measured by us. "-" denotes not available.

| Model | AIME24 | Math500 | AMC | Olymp | Average |
|---|---|---|---|---|---|
| 7B-scale SA models | | | | | |
| Qwen2.5-Math-7B-Instruct (Yang et al., 2024) | 13.3 | 79.8 | 50.6 | 40.7 | 46.1 |
| DeepSeek-R1-Qwen-7B$*$ (DeepSeek-AI, 2025) | 50.3 | 90.9 | 80.7 | 67.4 | 72.3 |
| GRESO-7B (DM) (Zheng et al., 2025) | 32.5 | 82.2 | 80.7 | 44.1 | 59.9 |
| GRESO-7B (OR1) (Zheng et al., 2025) | 35.0 | 82.3 | 64.5 | 45.7 | 56.9 |
| SimpleRL-Zoo-7B (Zeng et al., 2025) | 40.0 | 80.2 | 70.0 | 39.0 | 57.3 |
| PRIME-Zero-7B (Cui et al., 2025) | 16.7 | 83.8 | 62.7 | 40.9 | 51.0 |
| OpenReasoner-Zero-7B @8k (Hu et al., 2025) | 13.3 | 82.4 | 54.2 | 47.9 | 49.5 |
| Oat-Zero-7B (Liu et al., 2025b) | 43.3 | 80.0 | 62.7 | 41.0 | 56.8 |
| RLSC (Li et al., 2025) | 26.7 | 72.6 | 54.7 | 35.9 | 47.5 |
| Eurus-7B (Yuan et al., 2024) | 16.7 | 83.8 | 62.7 | 40.9 | 51.0 |
| GPG-7B (Chu et al., 2025) | 33.3 | 80.0 | 65.0 | 42.4 | 55.2 |
| TTRL-7B (Zuo et al., 2025) | 40.2 | 83.4 | 68.1 | - | 63.9 |
| Seed-GRPO (Chen et al., 2025b) | 50.0 | 91.6 | 78.3 | 61.5 | 70.4 |
| rStar-Math-7B (Guan et al., 2025) | 26.7 | 78.4 | 47.5 | 47.1 | 49.9 |
| Eurus-2-7B-PRIME (Cui et al., 2025) | 26.7 | 79.2 | 57.8 | 42.1 | 51.5 |
| Sky-T1-7B-Zero (Team, 2025) | 23.8 | 77.6 | 65.0 | 41.3 | 51.9 |
| Sky-T1-7B-RL (Team, 2025) | 24.6 | 85.6 | 69.0 | 49.3 | 57.1 |
| S1.1-7B (Muennighoff et al., 2025) | 19.2 | 82.0 | - | 43.1 | 48.1 |
| Bespoke-Stratos-7B (Labs, 2025) | 18.3 | 81.2 | - | 45.0 | 48.2 |
| 7B-scale SSM/SWA models | | | | | |
| PROMPTCOT-MAMBA-7B (Zhao et al., 2025) | 35.2 | 84.6 | - | 50.7 | 56.8 |
| SWARR-7B (Ours) | **54.1** | **94.6** | **82.4** | **71.6** | **75.7** |

During the RL stage, the learning rate is fixed at $10^{-6}$, with a weight decay of 0.1. Training runs for 400 steps, with one optimization per RL step. The rollout batch size is 128, and the number of RL groups is 8. The reward is 1 when the output strictly matches the answer. By default, we employ DAPO for RL training with token-level loss. RL training is conducted on a single node with 8 NVIDIA H800 GPUs. **(2) Strong training settings:** Building upon the fair comparison settings, we make the following adjustments for our strongest models. For 1.5B models, we extend the RL training steps from 400 to 1,400. For 7B models, we use the full AceReason dataset (math and code), increase the SFT batch size to 16, set the number of RL groups to 16 with a context length of 16k, change token-level loss to sequence-level loss for further stability, and extend the RL training steps to 600.

### 4.2 REASONING EVALUATION

We evaluate the reasoning performance of our proposed *SWARR* framework with other RL-based methods on Transformer, Hybrid, and Linear architectures at 1.5B and 7B scales. We highlight M1 (Wang et al., 2025b) and PROMPTCOT-MAMBA (Zhao et al., 2025) for comparison.

**Comparison with 1.5 RL reasoning models.** We compare our *SWARR-1.5B* against other RL algorithms that utilize Qwen2.5-Math-1.5B as the base model in Table 2. *SWARR-1.5B* achieves the highest average score (58.10), outperforming GRESO-1.5B (OR1), Oat-Zero-1.5B, and Seed-GRPO by 11.6%, 11.9% and 10.4%. Our *SWARR-1.5B* achieves higher performance than DeepSeek-R1-Distill-Qwen-1.5B by 1.9%. On AIME24, *SWARR-1.5B* achieves 27.7%, which is higher than Seed-GRPO and GRESO-1.5B (OR1). For Math500, *SWARR-1.5B* reaches 87.0%, surpassing all other methods, including GRESO-1.5B (DM) and Oat-Zero-1.5B. On AMC and Olympiad, *SWARR-1.5B* also leads with 61.4% and 56.1%. Notably, Our *SWARR-1.5B* outperform M1-3B (Wang et al., 2025b), a hybrid Transformer-Mamba reasoning models with $2\times$ parameters, by 2.8% in average score, demonstrating the effectiveness of our SWA approach even against larger models.

**Comparison with 7B RL reasoning models.** We presents a comprehensive comparison between our *SWARR-7B* model and other 7B-scale Transformers based on Qwen2.5-Math-7B in Table 3. Notably, our *SWARR-7B* outperforms DeepSeek-R1-Distill-Qwen-7B by 3.4%. We also include the

Table 4: Fair comparison of Self-Attention (SA) and Sliding-Window Attention (SWA) models at 1.5B scale on math-reasoning benchmarks. Both architectures are evaluated after supervised fine-tuning (SFT) and reinforcement learning (RL) under matched training budgets. Train/Eval time is reported in GPU hours.

| Model | Train Time | Eval Time | AIME24 | Math500 | AMC | Olymp. | Avg |
|-------|-----------|-----------|--------|---------|-----|--------|-----|
| *Fair Comparison after SFT* | | | | | | | |
| SA-1.5B-SFT | 306 | 3.72 | 18.85 | 79.50 | 49.10 | 47.18 | 48.66 |
| SWA-4k-1.5B-SFT | 249 | 1.27 | 16.25 | 78.20 | 47.29 | 47.33 | 47.27 |
| *Fair Comparison after RL* | | | | | | | |
| SA-1.5B-SFT-RL | 306+413 | 2.79 | 22.29 | 82.50 | 54.97 | 50.96 | 52.68 |
| SWA-4k-1.5B-SFT-RL | 249+386 | 0.91 | 22.81 | 85.70 | 57.53 | 51.78 | 54.46 |

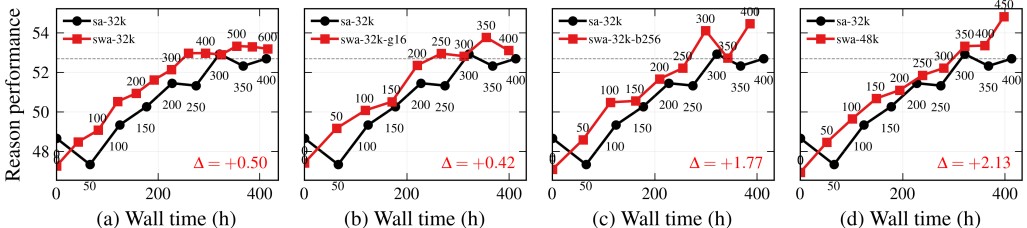

Figure 4: Performance comparison of SA and SWA under the same wall-clock time. (a) Under the same context length, SWA achieves 600 iterations while SA only completes 400. (b) SWA with a group size of 16 achieves higher performance than SA. (c) SWA with a batch size of 256 achieves higher performance than SA. (d) SWA with a longer context of 48k achieves higher performance than SA.

performance of PROMPTCOT-Mamba-7B (Zhao et al., 2025), a Mamba2 based reasoning model built upon Codestral (Jiang et al., 2023) and adapted for mathematical reasoning tasks. The results demonstrate that our *SWARR* approach surpasses the performance of PROMPTCOT-Mamba-7B by 18.9%, highlighting the effectiveness of *SWARR* for long-context reasoning in RL training.

### 4.3 FAIR COMPARISON OF SWA AND SA

To further validate the effectiveness of the SWA architecture, we conduct a fair comparison between SWA and SA models trained on the same dataset with identical training budgets. Both models are initialized from Qwen2.5-Math-1.5B and trained with the same time and memory budgets through both SFT and RL. The SWA model uses a window size of 4k, while the SA model is adapted to support a 32k context length. We present the results in Table 4 and visualize the performance under various training settings in Figure 4.

**SWA matches SA in SFT performance while being more efficient.** During the SFT stage, our SWA model with a 4k window (SWA-4k-1.5B-SFT) achieves an average accuracy of 47.27%, which is comparable to the 48.66% of the SA (SA-1.5B-SFT), as shown in Table 4. This demonstrates that the transition to SWA does not significantly compromise accuracy. Crucially, this comparable performance is achieved with greater efficiency: the SWA model trains 1.23× faster (249 vs. 306 GPU hours) and evaluates 2.9× faster (1.27 vs. 3.72 GPU hours) than its SA counterpart.

**SWA outperforms SA after RL.** After the RL stage, the SWA-4k-1.5B-SFT-RL model achieves an average score of 54.46% under same training time, surpassing the *SWARR-1.5B*'s score of 52.68% by 1.78%. This improvement is consistent across individual benchmarks, with SWA outperforming SA on Math500 (+3.2%), AMC (+2.56%), and AIME24 (+0.52%). These results highlight that SWA not only maintains strong performance but can leverage its efficiency to achieve superior results after RL optimization.

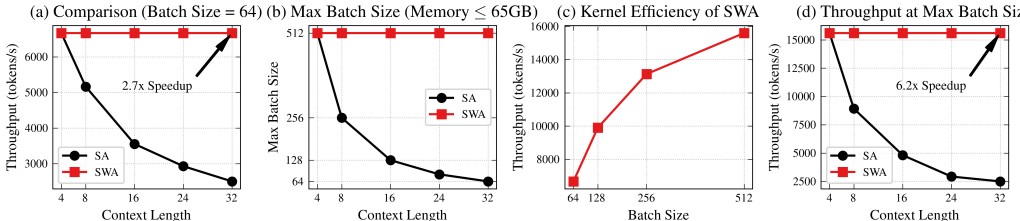

Figure 5: Performance comparison of Self-Attention (SA) and Sliding-Window Attention (SWA) models. (a) Throughput as a function of context length. (b) Maximum batch size supported under a 65GB memory constraint. (c) Effect of increasing batch size on throughput. (d) `SWA` achieves a 6.2× speedup over `SA` at 32k context length.

**`SWA` achieves greater gains by scaling batch, group, or context under same time budget.** The linear-time and constant-memory properties of `SWA` enable faster training steps with reduced memory usage. As shown in Figure 4(a), `SWA` completes 600 iterations compared to SA's 400 under identical 32k context and training time, leading to higher average reasoning accuracy. This efficiency allows SWA to scale (i) the number of DAPO groups, (ii) the mini-batch size, or (iii) the maximum context length, all within the same wall-clock time as the `SA` baseline. For instance, Figure 4(b) demonstrates that doubling DAPO groups from 8 to 16 incurs no latency penalty while yielding a 0.42% accuracy gain, highlighting the benefits of SWA's reduced memory footprint for policy updates. Similarly, Figure 4(c) shows a 1.77% performance gain when increasing the batch size from 128 to 256, with no additional time cost. Finally, Figure 4(d) illustrates that extending SWA's context to 48k tokens results in a 2.13% gains, confirming its linear scalability for long-context RL under fixed compute budgets.

## 4.4 ANALYSIS

**RL Training Time Breakdown.** Figure 7 illustrates the time breakdown of a single RL training step for both `SA` and `SWA` architectures. We show the time spent on Rollout, Logprob computation, and Training. We observe that the `SA` model with a 32k context length (`SA-32k`) is substantially slower, taking 526.5s per step, whereas the `SWA-4k` models are significantly more efficient, using just 287.6s. For `SWA` models, we present the average time with different window sizes, from 2k, 4k to 8k. As the window size increases, the total training time per step increases from 239.9s to 340s, primarily due to the increased computation in both Rollout and Training phases. Notably, for both `SA` and `SWA` models, the rollout process occupies nearly 87.31% (`SA-32k`) and 88.08% (`SWA-2k-32k`). We find that under the same context length, `SWA` requires only about half the rollout time compared to `SA`.

**Decoding Speed Comparison.** Decoding plays a critical role in RL rollouts, especially for long-context reasoning. We conduct a systematic evaluation to compare the efficiency of `SWA` and `SA` architectures in terms of memory consumption and inference latency. Figure 5 illustrates the scaling behavior of `SWA` and `SA` w.r.t. context length and batch size, revealing two key advantages of `SWA` over `SA`: (1) `SWA is computationally efficient`: As shown in Figure 5(a), `SWA` maintains high throughput regardless of context length, owing to its lower computational complexity. (2) `SWA is memory efficient`: Under a fixed memory constraint of 65GB, `SWA` supports a significantly larger maximum batch size than `SA`. As illustrated in Figure 5(b), the maximum batch size for `SA` decreases by a factor of 8 (from 512 to 64) as context length grows, while `SWA` remains largely unaffected. Figure 5(c) demonstrates that `SWA`'s throughput increases steadily with larger batch sizes. Furthermore, when comparing throughput at the maximum batch sizes (Figure 5(d)), `SWA` achieves a speedup of approximately 6.2×.

## 4.5 ABLATION STUDY

**SFT Convergence Speed Comparison.** We compare the loss curves during SFT for SA, `SWA`, and Liger (Lan et al., 2025) in Figure 6. All models are initialized from the same 4k pretrained Qwen2.5-Math-1.5B. The `SA` model is directly fine-tuned on 32k context, while `SWA` models are fine-tuned with 4k windows. Liger (Lan et al., 2025), which linearizes LLMs into gated recurrent structures, is

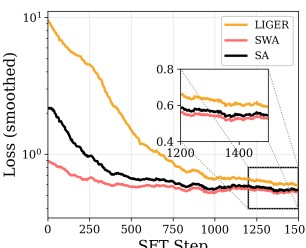
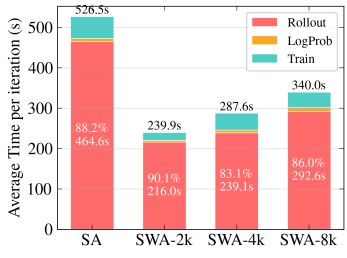
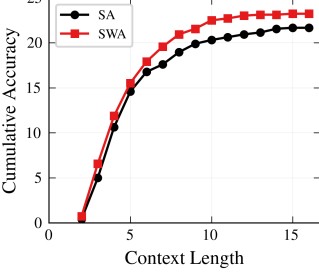

Figure 6: `SWA` converges faster than Liger and `SA` during SFT.

Figure 7: Time breakdown in RL Training for `SA` and SWA.

Figure 8: Generation Length vs. Cumulative Accuracy.

Table 5: Effect of supervised fine-tuning (SFT), RL, and window size on `SWA` and `SA` models. Train/Eval time is reported in GPU hours.

| Model | Train Time | Eval Time | AIME24 | Math500 | AMC | Olymp. | Avg |
|---|---|---|---|---|---|---|---|
| SA-1.5B-SFT | 306 | 3.72 | 18.85 | 79.50 | 49.10 | 47.18 | 48.66 |
| SFT Window Size Ablation | | | | | | | |
| SWA-2k-1.5B-SFT | 227 | 1.08 | 11.77 | 76.20 | 42.62 | 42.51 | 43.27 |
| SWA-4k-1.5B-SFT | 249 | 1.27 | 16.25 | 78.20 | 47.29 | 47.33 | 47.27 |
| SWA-8k-1.5B-SFT | 284 | 1.93 | 17.19 | 79.90 | 48.95 | 46.22 | 48.06 |
| RL Window Size Ablation | | | | | | | |
| SWA-2k-1.5B-SFT+RL (150 step) | 227+102 | 0.91 | 14.79 | 79.90 | 49.10 | 44.88 | 47.17 |
| SWA-4k-1.5B-SFT+RL (100 step) | 249+82 | 1.09 | 17.81 | 79.50 | 50.45 | 48.52 | 49.07 |
| SWA-8k-1.5B-SFT+RL (50 step) | 284+44 | 1.62 | 18.02 | 78.80 | 49.40 | 47.77 | 48.50 |

included for reference. We find that `SWA` achieves substantially faster loss reduction and lower final loss than both `SA` and Liger, highlighting its superior convergence speed and adaptation efficiency for long-context reasoning.

**Impact of Window Size of SWA.**  Window size directly affects model performance and computational efficiency. As shown in Table 5, increasing the window size from 2k to 4k leads to a substantial improvement of 4%, and further increasing to 8k yields a smaller gain of 0.79%. However, larger window size also incur higher training and evaluation times (e.g., 1.22 for 4k vs. 1.67 GPU hours for 8k). This demonstrates a clear trade-off: while larger windows can marginally improve performance, they come at the cost of increased computation. The 4k window size offers a better trade-off between performance and efficiency and is chosen as the default in our experiments.

**Impact of Generation Length.**  To assess the effect of generation length on model performance, we conduct an ablation study comparing `SWA` and `SA` architectures with generation lengths of 4k, 8k, and 16k tokens. For each setting, we measure the average performance on the four benchmarks. Figure 8 shows that cumulative accuracy rises quickly with context length, indicating the benefit of longer contexts for reasoning.

## 5 CONCLUSION

In this paper, we identify the quadratic complexity of self-attention (SA) as a key bottleneck for reinforcement learning (RL) training, particularly for long-context rollouts. We introduce Sliding-Window Attention for Reinforced Reasoning (SWARR), a two-stage framework that efficiently converts a pretrained `SA` transformer into an `SWA` model and further optimizes it using the DAPO algorithm. Under identical training conditions, our experiments show that *SWARR* matches or surpasses `SA` models on challenging mathematical reasoning benchmarks and outperforms Transformer and Hybrid models of comparable size. Importantly, *SWARR* delivers substantially faster rollout speeds and lower memory usage, enabling longer training context and more efficient utilization of computational resources.

# 6 ETHICS STATEMENT

This work does not involve human subjects, personally identifiable information, or sensitive data. All datasets used are publicly available and comply with relevant privacy and legal standards. The proposed methods are intended for research purposes in mathematical reasoning and do not promote harmful, discriminatory, or unethical applications. There are no known conflicts of interest or sponsorship issues related to this research. We have followed best practices for reproducibility and research integrity throughout the study.

# 7 REPRODUCIBILITY STATEMENT

We have taken comprehensive steps to ensure the reproducibility of our results. All experimental settings, model architectures, training procedures, and evaluation protocols are described in detail in the main paper, Section 4.1, and Appendix E. The datasets are publicly available on Hugging Face, as detailed in Section 4.1. We will release the code, model checkpoints, and detailed instructions to facilitate replication of our experiments upon publication.

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

## A  LIMITATION

Although our proposed *SWARR* has demonstrated promising results and great potential in RL training, several limitations and open questions remain:

**Domain Generalizability**  While our approach demonstrates strong efficiency and performance, its experimental validation is primarily confined to mathematical-reasoning benchmarks. As a result, the generalizability of SWA-based RNNs to other domains—such as open-ended dialogue, code generation, or tasks involving multimodal inputs—remains untested. Future work is needed to systematically evaluate the applicability and robustness of SWA in these broader contexts.

**Global Context Limitations**  Although we have shown that `SWA` expands the effective receptive field and can outperform `SA` under a fair computational budget, its effectiveness may be constrained on tasks that require truly global context. In particular, SWA may struggle with rare long-range dependencies that exceed the fixed window size, such as those encountered in long-context understanding or document-level reasoning tasks. This limitation is inherent to the local nature of the sliding-window mechanism.

**Potential Mitigations**  To address the above limitation, techniques such as multi-step inference or iterative context aggregation could be explored. For example, methods like Smooth Reading (Liu et al., 2025a) may help the model recover or approximate global context by sequentially processing overlapping windows or by leveraging external memory. Investigating such strategies is a promising direction for future research.

**Dependence on High-Quality Datasets**  The performance of *SWARR* is heavily reliant on the quality and scale of the datasets used for both SFT and RL. While we utilized extensive mathematical-reasoning datasets, any inherent biases or gaps in these datasets could limit the model's generalization to unseen problem types. Crafting high-quality datasets for complex reasoning remains a significant challenge.

## B  BROADER IMPACTS

The widespread adoption of *SWARR* as a potential successor to standard self-attention (`SA`) could catalyze a paradigm shift in the landscape of large language models. Its linear-time complexity and constant memory footprint during inference promise to significantly enhance efficiency, making large-scale deployment more feasible. This architectural advantage may fundamentally reshape modern inference engines such as vLLM (Kwon et al., 2023) and SGLang (Zheng et al., 2024), as the need for complex KV-cache management would be obviated, simplifying system design. Consequently, *SWARR* could pave the way for practical lifelong-serving scenarios, such as persistent personal assistants that handle continuous, long-running interactions. Furthermore, by alleviating the memory burden of the KV cache, *SWARR* unlocks new potential for on-device models, enabling them to achieve ultra-low latency and greater capabilities on edge devices.

## C  DISCUSSION

### C.1  CONCURRENT WORKS

Concurrent to our work, M1 (Wang et al., 2025b) explores reinforcement learning on a hybrid Mamba-Transformer architecture. Similar to other hybrid models such as Nemotron-H (Blakeman et al., 2025) and MiniMax-M1 (Chen et al., 2025a), M1 aims to balance performance and efficiency. A key contribution of M1 is its exploration of initializing the Mamba components with limited data, thereby avoiding costly training from scratch. This aligns with our motivation to efficiently adapt pretrained models to new architectures. However, a fundamental difference remains: M1 is a hybrid model that still contains quadratic self-attention layers, whereas our *SWARR* framework relies on a pure sliding-window attention mechanism, which offers strictly linear complexity. This distinction ensures that *SWARR* maintains a significant advantage in efficiency and scalability, particularly in scenarios involving extremely long contexts.

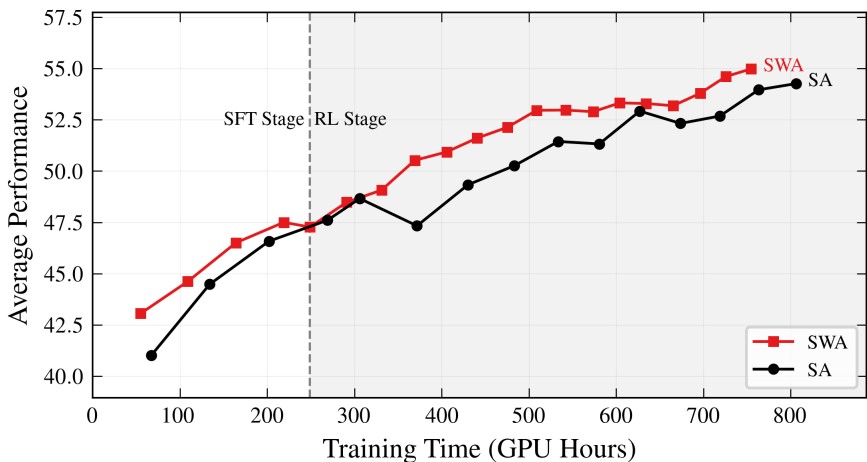

Figure 9: Illustration of the SWARR training pipeline. We present average performance across reasoning benchmarks. During the SFT stage, SWA achieves comparable results to SA. In the RL stage, SWA consistently outperforms SA.

Another relevant work is PROMPTCOT-MAMBA (Zhao et al., 2025), which investigates the reasoning capabilities of a pure Mamba2 architecture. The authors adapt a Mamba2 model, originally pretrained on coding data, for mathematical reasoning by applying supervised fine-tuning (SFT) on relevant corpora. Their work is valuable in demonstrating the potential of pure state-space models (SSMs) for reasoning tasks. In contrast, our work goes further: while they focus on SFT, we address the more complex challenge of reinforcement learning. We are the first to successfully apply RL to an `SWA` model for reasoning and to demonstrate that it can surpass the performance of its `SA` counterpart.

### C.2 NECESSITY OF THE TWO-STAGE TRAINING PIPELINE

Our framework is built upon a two-stage training pipeline: supervised fine-tuning (SFT) followed by reinforcement learning (RL). This design is crucial for successfully adapting a pretrained self-attention model to a sliding-window architecture for complex reasoning tasks. The SFT stage serves as an essential "cold start" for the model. The architectural shift from global attention (`SA`) to local attention (`SWA`) is significant, and our experiments show that directly applying RL to a freshly converted `SWA` model results in training instability and a failure to converge. The model's performance does not steadily improve, as the policy is not yet adapted to reasoning within a constrained local window.

Therefore, the SFT phase is indispensable. By fine-tuning on a large corpus of mathematical data, the model learns to effectively utilize its local receptive field, establishing a robust baseline policy. Once this foundation is in place, the RL stage can effectively and continuously refine the model's reasoning capabilities. As shown in Table 5, the SWA-4k-1.5B model after both SFT and RL achieves a significantly higher score (49.07) compared to the model with SFT alone (47.27). This demonstrates that while SFT provides the necessary initial capabilities, RL is key to unlocking the model's full potential, allowing it to surpass the performance of the SFT-only version. Furthermore, our results indicate that a window size of 4k provides a favorable trade-off between performance and computational efficiency.

### C.3 MORE DISCUSSIONS ON THE LOCALITY OF THOUGHT EXPERIMENTS

In our locality of thought motivation experiment, we also present the accumulated attention score distributions for both reasoning tasks and long-context understanding tasks. Notably, the locality phenomenon—where most attention is concentrated within a local window—holds strongly for reasoning tasks but does not generalize to long-context understanding tasks. This explains why

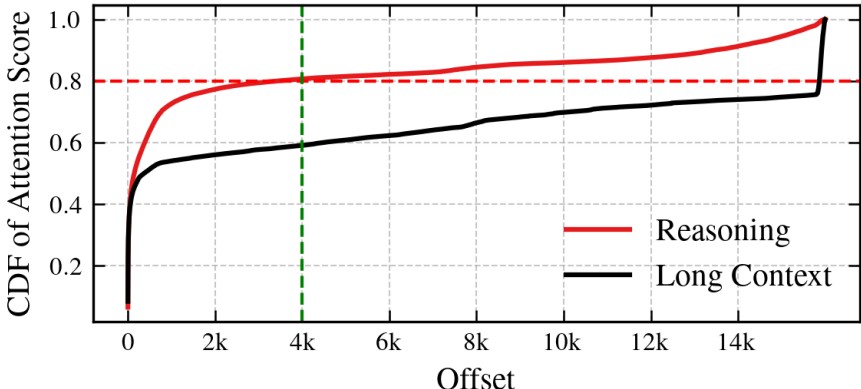

Figure 10: Comparison of accumulated attention scores for reasoning and long-context understanding tasks.

sliding-window attention (SWA) is highly effective for reasoning, yet may struggle with tasks requiring global context or long-range dependencies. As discussed in the Section A, techniques such as Smooth Reading (Liu et al., 2025a) could help mitigate this limitation by sequentially aggregating context or leveraging external memory, potentially extending the applicability of SWA to broader domains.

## D  DETAILS OF THE LOCALITY OF THOUGHT EXPERIMENTS

To quantitatively analyze the locality of attention, we define a metric for the accumulated attention score, following prior work on attention locality Prystawski et al. (2023); Wei et al. (2022). For a given sequence $\{x_i\}$, let $a_{ij}$ be the attention score between the query token $x_i$ and the key token $x_j$. The attention score of $x_i$ with an offset $d$ is denoted as $a_{i,i-d}$.

The accumulated attention score for a token $x_i$ up to an offset $d$, denoted as $A_{i,d}$, is the sum of attention scores from tokens within a window of size $d$:

$$A_{i,d} = \begin{cases} \sum_{k=0}^{d} a_{i,i-k} & d \leq i \\ 1 & d > i \end{cases} \tag{3}$$

By definition, $A_{i,d}$ is a monotonically increasing function of $d$ and is upper-bounded by 1.

To study the overall attention locality across different contexts, we average this score over tokens, layers, heads, and batches. The average accumulated attention score for a given offset $d$, $\bar{A}_d$, is calculated as:

$$\bar{A}_d = \frac{1}{B \cdot N_L \cdot N_H} \sum_{b=1}^{B} \sum_{l=1}^{N_L} \sum_{h=1}^{N_H} \left( \frac{1}{W} \sum_{i=L_{seq}-W+1}^{L_{seq}} A_{i,d} \right) \tag{4}$$

where $B$ is the batch size, $N_L$ is the number of layers, $N_H$ is the number of heads, and $L_{seq}$ is the sequence length.

To ensure a fair comparison and mitigate the influence of varying sequence lengths, we sample sequences with lengths between 1.5k and 1.6k tokens. We only average the accumulated attention scores over the last $W$ tokens of the generated response to minimize biases from response length variations. This formulation allows us to generate the Cumulative Distribution Function (CDF) of attention scores shown in Figure 1.

# E  DETAILS OF EXPERIMENTS

## E.1  TRAINING DATASET

**SFT Dataset.**  For supervised fine-tuning, we use `AceReason-1.1-SFT` (Liu et al., 2025c), a diverse, high-quality dataset focused on math and code reasoning. All responses in the dataset were generated by DeepSeek-R1 (DeepSeek-AI, 2025). It contains 2,668,741 math samples and 1,301,591 code samples, sourced from a wide range of datasets including OpenMathReasoning, NuminaMath-CoT, OpenCodeReasoning, MagicoderEvolInstruct, opc-sft-stage2, LeetCode, TACO, and APPS. To ensure the integrity of our evaluation, we perform rigorous data decontamination, filtering out any sample that has a 9-gram overlap with the test sets of our math and coding benchmarks. For the 1.5B models, we use only the math subset, whereas for the 7B models, we use both the math and code subsets. For more details, please refer to the AceReason technical report.

**RL Dataset.**  For reinforcement learning, we use `AceReason-Math`, a high-quality, verifiable, and challenging math dataset specifically curated for RL. This dataset contains 49,000 math problems and their corresponding answers, sourced from NuminaMath and DeepScaler-Preview. To ensure the suitability of the data for RL training, we applied stringent filtering rules to exclude problems with multiple sub-questions, multiple-choice or true/false formats, overly long and complex answers, proofs, or those requiring figures. This dataset was instrumental in training the AceReason-Nemotron models, which have demonstrated strong performance on difficult math benchmarks such as AIME24 and AIME25.

## E.2  EVALUATION BENCHMARKS

To provide a comprehensive assessment of our models' reasoning capabilities, we evaluate them on a diverse suite of challenging mathematical benchmarks.

**MATH**  The MATH dataset (Hendrycks et al., 2021) is a widely recognized benchmark for mathematical problem-solving. It comprises 12,500 problems sourced from American high-school mathematics competitions, including the AMC 10, AMC 12, and the AIME. The problems span seven distinct subjects—Pre-Algebra, Algebra, Number Theory, Counting & Probability, Geometry, Intermediate Algebra, and Precalculus—and are categorized into five difficulty levels. For our evaluation, we use the standard 500-problem test set, referred to as MATH500.

**AIME**  The American Invitational Mathematics Examination (AIME) is a prestigious and highly challenging mathematics competition for high-school students. It serves as a qualifier for the United States of America Mathematical Olympiad (USAMO). The problems on the AIME are significantly more difficult than those on the AMC and require a deeper level of mathematical insight and problem-solving skill. We use problems from the 2024 AIME competition (MAA, 2025) to test the advanced reasoning abilities of our models.

**AMC**  The American Mathematics Competitions (AMC) (MAA, 2023) are a series of nationwide contests in the United States designed to identify and nurture talent in mathematics. The competition series includes the AMC 10 (for students in grade 10 or below) and the AMC 12 (for students in grade 12 or below). The problems cover a wide range of high-school mathematics topics and are designed to be both challenging and accessible.

**OlympiadBench**  OlympiadBench (He et al., 2024) is a benchmark specifically designed to test the limits of mathematical reasoning at the level of international olympiads. It consists of problems from various national and international math olympiads, which are known for their extreme difficulty and the creative, non-standard solutions they often require. This benchmark serves as a stringent test of a model's ability to handle complex, multi-step reasoning and abstract mathematical concepts.

## E.3  TRAINING HYPERPARAMETERS

**SFT:**  The learning rate is initialized at $7 \times 10^{-6}$ and decayed to $7 \times 10^{-7}$ using a cosine-annealing schedule. Weight decay is set to 0.1. Training is performed on 4 nodes (32 GPUs in total).

**RL:** The RL phase uses a modified DAPO algorithm Yu et al. (2025a;b). The learning rate is fixed at $10^{-6}$, with weight decay of 0.1. Training runs for 1 600 steps, with one optimization per RL step. The rollout batch size is 128, and the number of RL groups is 8. Asymmetric clipping uses `h-clip` at 0.28 and `l-clip` at 0.2. The reward is based on `strict_match`, with an over-length penalty enabled. RL training is conducted on a single node with 8 NVIDIA H800 GPUs.

**Step 1: Initial Population**

**Ring Buffer (Size N=8)**

$KV_1$  $KV_2$  $KV_3$  $KV_4$

Write Pointer

**Step 2: Buffer is Full**

**Ring Buffer (Size N=8)**

$KV_1$  $KV_2$  $KV_3$  $KV_4$  $KV_5$  $KV_6$  $KV_7$  $KV_8$

Write Pointer

Wrap Around

**Step 3: Overwriting Oldest Data**

**Ring Buffer (Size N=8)**

$KV_9$  $KV_2$  $KV_3$  $KV_4$  $KV_5$  $KV_6$  $KV_7$  $KV_8$

Write Pointer

Figure 11: `SWA` ring-based buffer mechanism. The diagram illustrates the operational lifecycle of the fixed-size ring buffer for KV-cache management in three steps. (1) Initial population: the buffer is filled sequentially, and the attention window covers the active KV pairs. (2) Buffer full: once capacity is reached, the write pointer wraps around to the beginning of the buffer. (3) Overwriting oldest data: a new KV pair ($KV_9$) replaces the oldest entry ($KV_1$), ensuring the buffer always contains the most recent tokens while maintaining a constant memory footprint.

### E.4 DAPO MODIFICATIONS FOR SWARR

We adapt the DAPO algorithm Yu et al. (2025a;b) for `SWA` training with several targeted modifications to address the unique characteristics of sliding-window attention:

**Clip-Higher:** In contrast to standard DAPO, we employ strict on-policy training, which inherently mitigates the entropy-collapse issue that Clip-Higher was designed to address. The on-policy constraint ensures sufficient exploration, eliminating the need for asymmetric clipping ranges.

**Dynamic Sampling:** DAPO's dynamic sampling strategy typically resamples rollout batches whose rewards are all 0 or all 1 until the desired batch count is achieved. In our approach, we simply filter out such batches without resampling, streamlining the sampling process.

**Overlength Penalty:** We refine the overlength penalty mechanism by leveraging finish-reason information. The model may terminate generation because of a natural stop, reaching the length limit, or encountering an error. We explicitly encourage natural stops by applying a penalty to the reward when the finish reason is not as expected, discouraging truncation by length limits.

**KL-Divergence Removal:** Consistent with DAPO (Yu et al., 2025b), we remove the KL-divergence term from the GRPO objective. This simplification is especially advantageous for `SWA`

training, as it enables the model to diverge more freely from the initial policy and better adapt to the constrained attention pattern.

## F    DETAILS OF THE INFRASTRUCTURE

**Ring-based Cache Management for SWA:**    The `SWA` cache-management strategy is visualized in Figure 11, which details the operation of the ring-based buffer. Initially, the buffer of size $N$ is populated sequentially with key–value (KV) pairs corresponding to incoming tokens. As the system processes the sequence, the attention window of size $K$ slides to cover the most recent entries. Once the buffer's capacity is reached, its circular nature is activated: the write pointer wraps around from the end to the beginning. The core overwrite mechanism is shown when the next KV pair ($KV_9$) arrives; it replaces the oldest data ($KV_1$) in the buffer. This process effectively implements a circular queue, ensuring that the cache always holds the $K$ most recent KV pairs within a fixed memory allocation. This design maintains a constant memory footprint and facilitates efficient, constant-time read/write operations, eliminating the overhead associated with traditional dynamic KV-cache management for long sequences.

**Distributed RL Training Framework**    Our RL training pipeline is built upon the XTuner (Contributors, 2023) framework, which is designed for efficient fine-tuning of large language models. To handle the substantial memory requirements of RL training, we employ Fully Sharded Data Parallel (FSDP) (Zhao et al., 2023), which shards model parameters, gradients, and optimizer states across multiple GPUs. This allows us to train large models that would otherwise not fit into a single device's memory. Furthermore, to minimize the overhead associated with switching between training and inference (rollout), we adopt a colocation strategy inspired by HybridFlow (Sheng et al., 2024). This approach allows both training and inference processes to run concurrently on the same set of devices, eliminating the need for costly weight transfers and conversions between different formats, thereby accelerating the overall RL loop.

**Optimized Inference Engine for Rollout**    The efficiency of the rollout phase is critical for RL training throughput. Our inference engine is heavily optimized to maximize performance. It features a centralized request distributor that manages incoming generation requests and batches them dynamically. We implement continuous batching (Yu & Jeong, 2022), which allows new requests to be added to a running batch, improving GPU utilization by avoiding idle time between batches. Additionally, we use asynchronous rollout (Fu et al., 2025), where the generation of sequences is decoupled from the main training loop, allowing the model to continuously produce experiences without blocking the policy optimization step. These optimizations work in concert to ensure a high-throughput, low-latency rollout process, which is essential for effective and scalable RL training.

## G    DETAILS OF RELATED WORKS

In this section, we provide a more detailed discussion of related works on efficient attention mechanisms and their application to reinforcement learning for reasoning tasks. We expand on the comparison presented in the main paper, categorizing methods by their architectural approach and evaluating them across multiple dimensions, including computational complexity, memory usage, ease of conversion from pretrained self-attention models, performance in RL settings, and training costs.

### G.1    EFFICIENT ATTENTION MECHANISMS

#### G.1.1    SELF-ATTENTION AND ITS VARIANTS

Standard self-attention (Vaswani et al., 2017) remains the cornerstone of modern Transformers. By materializing an $L \times L$ score matrix, it offers a global receptive field, yet its $O(L^2)$ time and memory footprint quickly becomes the bottleneck when long contexts or many roll-outs are required, e.g., in reasoning-oriented RL systems such as DeepSeek-R1 (DeepSeek-AI, 2025).

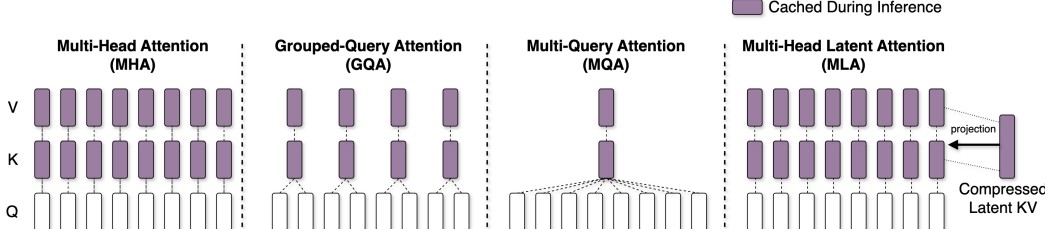

Figure 12: Comparison of self-attention variants in terms of memory and computational efficiency.

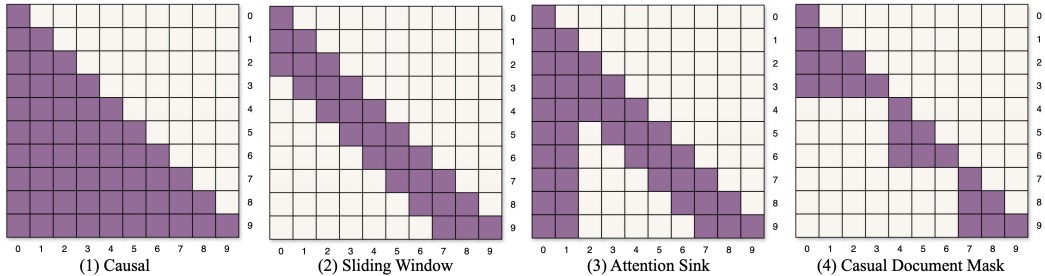

Figure 13: Attention mask patterns in Sparse Attention. (1) Causal, (2) Sliding Window, (3) Attention Sink, (4) Causal with Document Mask.

**Multi-Head Attention (MHA)** As illustrated in Figure 12, MHA (Vaswani et al., 2017) employs a separate key (K) and value (V) head for each query (Q) head. This design parallelizes $h$ independent attention heads, each operating on a fraction of the model's dimensions ($d_{qk} = d_{\text{model}}/h$). While this allows the model to jointly attend to information from different representation subspaces, it is computationally expensive. The KV-cache, which stores the key and value matrices, scales linearly with the number of heads, consuming $O(h \cdot L \cdot d_{\text{model}})$ space. The time complexity remains quadratic at $O(L^2 \cdot d_{\text{model}})$ due to the full attention matrix calculation, making it a bottleneck for long sequences.

**Grouped-Query Attention (GQA)** GQA (Ainslie et al., 2023) offers a compromise between MHA and MQA by grouping query heads to share a single key and value head. As shown in the figure, this reduces the number of K and V heads from $h$ to $g$ (where $g < h$). This significantly reduces the size of the KV-cache to $O(g \cdot L \cdot d_{\text{model}})$, leading to lower memory usage and faster inference. The time complexity is still dominated by the quadratic attention calculation, but the constant factor is reduced.

**Multi-Query Attention (MQA)** MQA (Shazeer, 2019) represents a more aggressive optimization where all query heads share a single key and value head. This drastically reduces the KV-cache size to $O(L \cdot d_{\text{model}})$, making it the most memory-efficient variant among MHA and GQA. While the computational complexity remains quadratic at $O(L^2 \cdot d_{\text{model}})$, the reduction in memory bandwidth requirements for loading the keys and values leads to significant speedups in practice, especially during autoregressive decoding. However, this aggressive sharing can sometimes lead to a drop in model quality due to the reduced capacity of the attention mechanism.

**Multi-head Latent Attention (MLA)** MLA (DeepSeek-AI et al., 2024) introduces a further optimization by compressing the key and value pairs into a shared, low-rank latent space before the attention operation. As depicted in the figure, this involves a projection to a "Compressed Latent KV." This reduces the memory footprint of the KV-cache from $O(L \cdot d_{\text{model}})$ to $O(L \cdot r)$, where $r$ is the rank of the latent space and $r \ll d_{\text{model}}$. While an extra matrix multiplication is introduced, the overall time complexity remains $O(L^2 \cdot d_{\text{model}})$, but with a smaller constant factor, especially for long sequences. This makes MLA a drop-in replacement for MHA that can significantly improve efficiency with minimal impact on performance.

### G.1.2 SPARSE ATTENTION PATTERNS

Figure 14: Recurrent update rules for linear-attention mechanisms: (a) gated vector recurrence, (b) optimization-based update, (c) outer-product state with decay, (d) delta-rule controlled forgetting. Each panel shows state update and readout for a representative architecture.

This section details various sparse attention mechanisms, many of which are supported by libraries such as FlashAttention (Dao et al., 2022). Figure 13 illustrates four common patterns.

**(1) Causal Attention** This is the standard attention mechanism in autoregressive models. Each token can attend to itself and all preceding tokens, as shown in Figure 13(1). Its computational complexity is $\mathcal{O}(n^2)$, where $n$ is the sequence length. The memory complexity is $\mathcal{O}(n^2)$ to store the attention matrix during training and $\mathcal{O}(n \cdot d)$ for the KV-cache during inference, where $d$ is the model dimension.

**(2) Sliding-Window Attention (SWA)** SWA, depicted in Figure 13(2), restricts each query to attend only to keys within a local window of fixed size $w$. This reduces the computational complexity from $\mathcal{O}(n^2)$ to $\mathcal{O}(nw)$ (Beltagy et al., 2020), with a corresponding memory complexity of $\mathcal{O}(nw)$ for the attention matrix. During inference, a ring-buffer KV-cache can maintain a constant memory footprint of $\mathcal{O}(w \cdot d)$. While this pattern is static and requires no additional training, the rigid locality bias can limit the capture of long-range dependencies.

**(3) Attention Sinks** As observed by StreamingLLM (Xiao et al., 2023), keeping the first few "sink" tokens plus the most recent window of tokens is often sufficient to maintain attention quality. This pattern is shown in Figure 13(3). It caches $s$ sink tokens and a sliding window of $w$ recent tokens, resulting in a computational complexity of $\mathcal{O}(n(s + w))$ for training. During inference, the

time per token is constant, and the KV-cache size is fixed at $\mathcal{O}((s + w) \cdot d)$, independent of the total sequence length.

**(4) Causal with Document Mask** This pattern (Figure 13(4)) is a variant of causal attention in which attention is restricted to certain blocks or segments, such as documents in a long context. This can be useful for tasks involving multiple documents, preventing cross-document attention while maintaining causality within each document. The computational and memory complexity depends on the specific block structure; for $k$ documents of average length $n/k$, the complexity is approximately $k \cdot \mathcal{O}((n/k)^2) = \mathcal{O}(n^2/k)$, offering a significant reduction if $k$ is large.

### G.1.3 LINEAR ATTENTION

Recent advances in efficient attention mechanisms have produced models expressible as recurrent neural networks (RNNs). These achieve linear-time complexity and constant memory at inference, suiting long sequences. Figure 14 group them by state-update mechanisms:

Table 6: Recurrent update rules for efficient architectures. $d$ is the hidden-state dimension.

| Model | Update rule ($S_t$) | Read-out ($o_t$) | Complexity |
|---|---|---|---|
| Vector-valued hidden state (classical/gated RNNs) | | | |
| HGRN (Qin et al., 2022) | $h_t = \alpha_t \odot h_{t-1} + (1 - \alpha_t) \odot v_t$ | $o_t = h_t \odot q_t$ | $O(d)$ |
| Hawk (RG-LRU) (Deo et al., 2024) | $h_t = r_t h_{t-1} + i_t \odot x_t$ | $o_t = h_t \odot q_t$ | $O(d)$ |
| Optimization-based update | | | |
| TTT (Sun et al., 2024a) | $S_t = S_{t-1} - \eta_t \nabla_{S_{t-1}} \mathcal{L}(S_{t-1}; x_t)$ | $o_t = S_t q_t$ | $O(d^2)$ |
| Matrix-valued state via outer products | | | |
| Linear Attention (Katharopoulos et al., 2020) | $S_t = S_{t-1} + v_t k_t^\top$ | $o_t = S_t q_t$ | $O(d^2)$ |
| + Kernel | $S_t = S_{t-1} + v_t \phi(k_t)^\top$ | $o_t = S_t \phi(q_t)$ | $O(d^2)$ |
| + Normalization | $S_t = S_{t-1} + v_t \phi(k_t)^\top, \quad z_t = z_{t-1} + \phi(k_t)$ | $o_t = S_t \phi(q_t)/(z_t^\top \phi(q_t))$ | $O(d^2)$ |
| RetNet/Lightning (Sun et al., 2023; 2024b) | $S_t = \gamma S_{t-1} + v_t k_t^\top$ | $o_t = S_t q_t$ | $O(d^2)$ |
| GLA (Yang et al., 2023) | $S_t = S_{t-1} \odot (\mathbf{1}\alpha_t^\top) + v_t k_t^\top$ | $o_t = S_t q_t$ | $O(d^2)$ |
| Mamba-2 (Dao & Gu, 2024) | $S_t = \gamma_t S_{t-1} + v_t k_t^\top$ | $o_t = S_t q_t$ | $O(d^2)$ |
| RWKV-6 (Peng et al., 2024) | $S_t = S_{t-1}\text{Diag}(\alpha_t) + v_t k_t^\top$ | $o_t = (S_{t-1} + (d \odot v_t)k_t^\top)q_t$ | $O(d^2)$ |
| HGRN-2/MetaLA (Qin et al., 2024b; Zhang et al., 2024) | $S_t = S_{t-1}\text{Diag}(\alpha_t) + v_t(1 - \alpha_t)^\top$ | $o_t = S_t q_t$ | $O(d^2)$ |
| Delta-rule / controlled-forgetting family | | | |
| Longhorn (Liu et al., 2024) | $S_t = S_{t-1}(\mathbf{I} - \frac{\beta_t k_t k_t^\top}{1 + \beta_t k_t^\top k_t}) + \frac{\beta_t v_t k_t^\top}{1 + \beta_t k_t^\top k_t}$ | $o_t = S_t q_t$ | $O(d^2)$ |
| DeltaNet (Li et al., 2023) | $S_t = S_{t-1}(\mathbf{I} - \beta_t k_t k_t^\top) + \beta_t v_t k_t^\top$ | $o_t = S_t q_t$ | $O(d^2)$ |
| Gated DeltaNet (Li et al., 2023) | $S_t = \alpha_t S_{t-1}(\mathbf{I} - \beta_t k_t k_t^\top) + \beta_t v_t k_t^\top$ | $o_t = S_t q_t$ | $O(d^2)$ |

**Vector-valued Hidden State (Classical/Gated RNNs)** This family includes models like HGRN (Qin et al., 2022) and Hawk (Deo et al., 2024), which use a vector-valued hidden state updated through gating mechanisms. The update rule is typically of the form $h_t = \alpha_t \odot h_{t-1} + (1 - \alpha_t) \odot v_t$, where $\alpha_t$ is a data-dependent gating vector. These models are reminiscent of classical RNNs but with modern architectural designs that allow for efficient, parallelizable training.

**Optimization-based Update** This family of models, including TTT (Sun et al., 2024a), formulates the state update as an optimization process. The state $S_t$ is updated by taking a gradient descent step on a local loss function $\mathcal{L}(S_{t-1}; x_t)$, as shown in the update rule $S_t = S_{t-1} - \eta_t \nabla_{S_{t-1}} \mathcal{L}(S_{t-1}; x_t)$. This perspective frames sequence modeling as a continuous optimization problem, where the model's state evolves to minimize a loss at each timestep.

**Matrix-valued State via Outer Products** This family, which includes RetNet/Lightning (Sun et al., 2023; 2024b), GLA (Yang et al., 2023), Mamba-2 (Dao & Gu, 2024), RWKV-6 (Peng et al., 2024), and HGRN-2/MetaLA (Qin et al., 2024b; Zhang et al., 2024), maintains a matrix-valued state $S_t$ that is updated via outer products. The general update rule is $S_t = \gamma S_{t-1} + v_t k_t^\top$, where $\gamma$ can be a scalar or a diagonal matrix, combining a decay factor with a rank-1 update. This formulation allows for parallel training like standard transformers while enabling recurrent inference.

**Delta-Rule / Controlled-Forgetting Family** This family, featuring DeltaNet (Li et al., 2023) and Gated DeltaNet (Li et al., 2023), employs a "controlled forgetting" mechanism inspired by the delta learning rule. The state is updated by selectively erasing information along a specific direction

before writing new information. The update rule is $S_t = S_{t-1}(\mathbf{I} - \beta_t k_t k_t^\top) + \beta_t v_t k_t^\top$, where $\beta_t$ controls the forgetting and writing rates. Gated versions add an additional gating term $\alpha_t$.

While these recurrent formulations offer significant efficiency gains, adapting pre-trained self-attention models to these architectures is often non-trivial, typically requiring complex distillation processes or training from scratch. This has limited their application in RL, where leveraging large pre-trained models is crucial.

Our SWA approach stands out for its balance of efficiency, ease of conversion, and strong RL performance, making it particularly suitable for practical deployment in reasoning tasks where pretrained models need adaptation without extensive retraining.

## H    USE OF LLMS

Large language models (LLMs) were used to polish, grammar-check, and refine the language to improve the readability of this paper. The core ideas, experimental design, and results were developed without the use of LLMs.

